# Exploring the Role of GITR/GITRL Signaling: From Liver Disease to Hepatocellular Carcinoma

**DOI:** 10.3390/cancers16142609

**Published:** 2024-07-22

**Authors:** Stavros P. Papadakos, Elena Chatzikalil, Georgios Vakadaris, Lampros Reppas, Konstantinos Arvanitakis, Theocharis Koufakis, Spyros I. Siakavellas, Spilios Manolakopoulos, Georgios Germanidis, Stamatios Theocharis

**Affiliations:** 1First Department of Pathology, School of Medicine, National and Kapodistrian University of Athens, 11527 Athens, Greece; stpap@med.uoa.gr (S.P.P.); elenachatz@med.uoa.gr (E.C.); 2First Department of Internal Medicine, AHEPA University Hospital, Aristotle University of Thessaloniki, 54636 Thessaloniki, Greece; vakadarisgeorgios@gmail.com (G.V.); arvanitak@auth.gr (K.A.); 3Basic and Translational Research Unit (BTRU), Special Unit for Biomedical Research and Education (BRESU), Faculty of Health Sciences, School of Medicine, Aristotle University of Thessaloniki, 54636 Thessaloniki, Greece; 44th Department of Internal Medicine, Attikon University Hospital, National and Kapodistrian University of Athens Medical School, 11527 Athens, Greece; lreppas@med.uoa.gr; 52nd Propedeutic Department of Internal Medicine, Aristotle University of Thessaloniki, Hippokration General Hospital, 54642 Thessaloniki, Greece; thkoyfak@hotmail.com; 62nd Academic Department of Internal Medicine, Liver-GI Unit, General Hospital of Athens “Hippocration”, National and Kapodistrian University of Athens, 114 Vas. Sofias str, 11527 Athens, Greece; s.siakavellas@gmail.com (S.I.S.); smanolak@med.uoa.gr (S.M.)

**Keywords:** cancer, immunotherapy, hepatocellular carcinoma (HCC), glucocorticoid-induced TNFR-related protein (GITR), anti-GITR monoclonal antibodies

## Abstract

**Simple Summary:**

Glucocorticoid-induced TNFR-related protein (GITR) is widely expressed in immune cells, mainly Tregs, enhancing their activation and proliferation. Pre-clinical evidence on using GITR agonists as a concomitant treatment to available antitumor immunotherapeutic options has demonstrated promising results. Hepatocellular carcinoma, despite being widely studied due to its high frequency among liver cancers, is a disease with high rates of recurrence and metastasis. During the last decades, investigational efforts have focused on specific target antigens for the development of immunotherapy strategies. As a part of this research, evaluating the potential role of novel immune targets is considered of great importance in terms of offering choices to patients with advanced or recurrent disease.

**Abstract:**

Hepatocellular carcinoma (HCC) is the most common primary liver cancer and presents a continuously growing incidence and high mortality rates worldwide. Besides advances in diagnosis and promising results of pre-clinical studies, established curative therapeutic options for HCC are not currently available. Recent progress in understanding the tumor microenvironment (TME) interactions has turned the scientific interest to immunotherapy, revolutionizing the treatment of patients with advanced HCC. However, the limited number of HCC patients who benefit from current immunotherapeutic options creates the need to explore novel targets associated with improved patient response rates and potentially establish them as a part of novel combinatorial treatment options. Glucocorticoid-induced TNFR-related protein (GITR) belongs to the TNFR superfamily (TNFRSF) and promotes CD8^+^ and CD4^+^ effector T-cell function with simultaneous inhibition of Tregs function, when activated by its ligand, GITRL. GITR is currently considered a potential immunotherapy target in various kinds of neoplasms, especially with the concomitant use of programmed cell-death protein-1 (PD-1) blockade. Regarding liver disease, a high GITR expression in liver progenitor cells has been observed, associated with impaired hepatocyte differentiation, and decreased progenitor cell-mediated liver regeneration. Considering real-world data proving its anti-tumor effect and recently published evidence in pre-clinical models proving its involvement in pre-cancerous liver disease, the idea of its inclusion in HCC therapeutic options theoretically arises. In this review, we aim to summarize the current evidence supporting targeting GITR/GITRL signaling as a potential treatment strategy for advanced HCC.

## 1. Introduction

Liver cancer represents the sixth most commonly diagnosed cancer globally and is the fourth leading cause of cancer-related mortality [1,2,3]. HCC is the predominant form of primary liver cancer, accounting for 75–85% of cases and posing a significant global health challenge [1,2]. Chronic liver disease, mainly cirrhosis, constitutes a precursor for most HCC cases, while hepatitis B virus (HBV), hepatitis C virus (HCV), aflatoxin, heavy alcohol intake and type 2 diabetes are considered the main risk factors associated with HCC development [4,5]. Moreover, inactivation of several tumor suppressor genes (e.g., p53), abnormal activation of oncogenes (e.g., KRAS) and various signaling pathways (e.g., PI3K/Akt, MAPK, JAK/STAT, Wnt/β-catenin), as well as dysregulation of epigenetic events (e.g., microRNAs), represent pre-tumorigenic activity involved in HCC development and progression, associated in many cases with disease prognosis [6,7]. An important aspect of the current investigation is also addressing potential ancillary biomarkers associated with HCC severity [8,9]. In addition to these genetic markers, HCC immunological tumor microenvironment (TME) has attracted extensive attention during the last decade [10]. It has been proved that the crosstalk between tumor cells and immune TME affects HCC progression by enhancing cell proliferation, survival, migration and invasion capacity [10,11]. A better understanding of these TME- associated mechanisms has led during the last few years to the development of novel therapeutic approaches as a part of HCC treatment options [10,11]. Immunotherapeutic agents, including vaccines, antibodies, immune checkpoint inhibitors and CAR-T therapy, are potential therapeutic choices that remain currently under investigation [10].

However, despite the aforementioned improvements in HCC immunotherapy, the results are currently considered unsatisfactory, even when it is combined with the other treatment options (specifically, chemotherapy, surgery, and radiofrequency ablation), with HCC patients presenting a high incidence of recurrence and metastasis [12,13,14,15,16,17,18]. Sorafenib is the most commonly used therapeutic option for HCC treatment, although it only offers a minimal enhancement of survival by 7–10 months [19]. Various combination therapies have also been explored, particularly including immune checkpoint inhibitors (ICIs) and VEGF inhibitors [20]. The IMbrave150 study with atezolizumab and bevacizumab showed improved overall survival (OS) and progression-free survival (PFS) over sorafenib, leading to its approval [21]. Similarly, the ORIENT-32 study with sintilimab and a bevacizumab biosimilar presented promising results [22]. Interestingly, combining CTLA-4 and PD-1/PD-L1 inhibitors has shown potential in increasing therapeutic activity. Ipilimumab and nivolumab demonstrated promising results [23], while the HIMALAYA study showed improved OS with a combination of tremelimumab and durvalumab [24]. Despite this progress, none of the above options has demonstrated a significant improvement in long-term survival compared to sorafenib monotherapy [10]. Moreover, although early diagnosis methods have been updated by the establishment of screening programs 13% of the patients still experience a diagnostic delay of a time interval above 3 months between initial presentation to diagnosis [25]. In order to address these unmet needs, research has focused on identifying mechanisms involved in HCC development that could be used as main or ancillary therapeutic targets, representing, in many cases, potential prognostic tools as well.

Glucocorticoid-induced tumor necrosis factor receptor-related protein (GITR) represents a potential immunotherapy target in various kinds of neoplasms (e.g., breast cancer, colorectal cancer), especially with concomitant use of PD-1 blockade [26]. Furthermore, pre-clinical evidence on GITR-agonist monotherapy has demonstrated antitumor effect via enhancing CD8^+^ and CD4^+^ effector T-cell activity and depleting tumor-infiltrating Tregs, which are considered the main GITR expressors [26]. Considering these data, we aim, in this review, to summarize current knowledge regarding GITR involvement in HCC development and to analyze its potential effect as a novel therapeutic target for HCC.

## 2. Overview of GITR/GITRL: Structure, Tissue Distribution, Signaling Pathways, and Functional Insights

Apoptosis, or programmed cell death, involves various molecules and takes place at the surface of the cells via ligand-receptor interactions. It is considered to be involved in cell and tissue development and in neoplastic processes as well [27,28,29]. Cysteine-dependent aspartate-specific proteases (caspases) regulate the apoptotic cascade, although cases of caspase-independent apoptosis have also been described [30,31]. Apoptosis occurs via two different pathways, both of which result in the activation of the caspase complex: the intrinsic and the extrinsic pathways [29,32]. The intrinsic pathway is regulated by B-cell lymphoma 2 (Bcl-2) proteins. Specifically, its activation is a result of mitochondrial inhibition, which leads to cytochrome c release [33]. The extrinsic pathway is regulated by death receptors, including tumor necrosis factors receptors (TNF-TNFR1), Fas ligand (FasL-Fas), and TNF-related apoptosis inducing-ligand [34,35]. TNFR superfamily includes three types of receptors: (1) costimulatory receptors, which activate nuclear factor kappa-light-chain enhancer (NF-κB), *p38* and *JNK2* pathways, (2) death domain receptors, which activate caspase 8 or NF-κB, and calf intestinal alkaline phosphatase (cIAP), and (3) decoy receptors, which inhibit ligands’ functionality [36]. Death domain receptors consist of cysteine-rich extracellular domains and an intracellular cytoplasmic region termed the ‘death domain’ (Figure 1A) [37].

The TNF/TNFR superfamily includes several molecules that have been proved to regulate cell death, proliferation, and differentiation [38,39]. Specifically, two TNFRs (TNFRI, TNFRII), the lymphotoxin α2β receptor (LTβR), the low-affinity nerve growth factor (NGF) receptor (NGFR), the lymphoid molecules (CD40, CD27, CD30, OX40,4-1BB) and the apoptosis receptors (Fas and DR3) are a part of the TNF/TNFR superfamily, presenting the typical structure of type I transmembrane proteins, as they contain a variety of 40 amino-acid-cysteine rich regions in their extracellular domain [37,38,39,40]. Their function presents heterogeneity in regulating cell proliferation and death signals [37]. TNFR expression levels present a great variety as well. For example, TNFR1 presents low expression levels in several types of cells, while CD27, TNFR1 and TNFR2 are mildly expressed specifically in naïve CD4 T cells [41,42]. In 1997, investigating this heterogenous pattern of structure and expression, and specifically, comparing untreated and dexamethasone-treated murine T cell hybridoma cells, a new gene was cloned, termed ‘*GITR*’. GITR stands for glucocorticoid-induced tumor necrosis factor receptor family-related gene and encodes a new protein of the TNF/TNFR family, selectively activated by the ‘*GITRL*’ ligand [37,43].

GITR gene (*GITR*) consists of 5 exons (Figure 1B) [43]. The first three exons encode the extracellular domain, exon 4 partially encodes the extracellular domain, the transmembrane domain and a part of the cytoplasmic domain, and exon 5 encodes the cytoplasmic domain [43]. *GITR* encodes three soluble products in mouse models and humans: mGITRD, mGITRD2 and hGITR, which may function as a decoy receptor, blocking GITR-GITRL interaction [43,44,45]. GITRL gene (*GITRL*), on the other hand, consists of 3 exons (Figure 1B), encoding a ligand consisting of 173 amino acids and presenting type II membrane protein features [46]. The extracellular domain of GITR is formed by three cysteine-rich domains (CRDs: CRD1, CRD2, CRD3) and a cytoplasmic domain, which presents good homology with the cytoplasmic domains of OX40, 4-1BB and CD27, and a quite good homology with CD40, evidence which, combined to the functional similarity, established the new GITR/GITRL subfamily as a part of TNFR superfamily [43,47]. GITR’s CRDs consist of 19 cysteines in 137 amino acids [47]. CDR1 consists of an A1 region, which is linked with CRD2 by a loop containing disulfide bonds to both the CRD1 A1 subunit and CRD2 A2 subunits. CDR2 consists of A2 and B1 regions, interacting, however, with GITRL only via the B1 region [47]. CRD3 contains a B1 region with a close to membrane receptor–receptor region, which creates receptor homodimers between different ligand–receptor complexes [47]. GITR is theoretically a trimer molecule; however, it may rarely present as a monomer or as different multimeric molecules [48].

Nocentini et al., who introduced GITR, reported *GITR* mRNA low-level expression in T cells, e.g., T cell hybridoma, thymocytes and peripheral T lymphocytes, positively correlated with T lymphocyte activation [37]. In their first study, mRNA expression was not detected in other tissues, specifically the liver, brain, and kidney; however, the authors did not exclude its existence, suggesting that *GITR* expression could potentially be the result of inflammatory processes and tissue regeneration [37]. Considering the fact that, in this study GITR expression level increased slowly after T cell activation or treatment with dexamethasone, the authors suggested that GITR activation is an indirect mechanism in the tissues studied [43]. Studies that followed indicated that GITR is expressed at various levels in many types of immune cells, specifically in antigen-presenting cells (APCs), including myeloid dendritic cells (DCs), T-cells, including T-regulatory cells (Tregs), B-cells, natural killer cells (NK cells) and macrophages (Figure 1) [46,49,50,51,52,53]. High levels of GITR expression have been reported in endothelial cells as well [43]. Regarding GITRL expression, it presents various levels in the different cell types, being highly expressed in endothelial cells, especially in microvascular-derived endothelial cells, and lowly expressed in B-cells and macrophages [54]. It is worth notable that in response to pro-inflammatory signals, GITRL expression is rapidly upregulated in APCs and ECs and decreases in 24–48 h [54]. Moreover, not all proinflammatory stimuli enhance the increase of GITRL expression levels: characteristically, in endothelial cells, GITRL expression is increased by interferons (IFNs) and not by proinflammatory cytokines [43].

TNFR members generally bind TNF ligands and activate the transcription NF-κB pathways by TRAFs, which further transfer signals within the cell [55]. Following this general pattern, after being activated, GITR binds to several TRAFs via a motif contained in the CRD2, subsequently activating NF-κB [56]. GITR also binds the protein arginine N-methyltransferase (PRMT1), enhancing type-I methylation reactions, being involved in protein trafficking, signaling pathways and transcriptional regulation, and is considered a potential activator of NF-κΒ. NF-κB-mediated GITR signaling blocks T-cell apoptosis and promotes T-cell survival via *BclxL* upregulation (Figure 1C) [56]. In the periphery, GITR/GITRL increases T-cell activation and CD25 expression cytokine production, enhancing T-cell proliferation (Figure 1C) [20]. Moreover, GITR increases CD8^+^ T-cell cytotoxicity and promotes the survival of bone marrow CD8^+^ memory T-cells [20]. GITR’s short-term stimulation decreases Treg’s ability to suppress effector T-cells, while its long-term stimulation promotes Treg functions (Figure 1C) [57]. Regarding GITR/GITRL effect on other types of immune cells, increased cell activation and antibody production in B-cells, enhancement of cell migration and pro-inflammatory cytokines release in DCs, decreased cytotoxicity and IFN-γ production in NK-cells, increase of inflammatory mediators and mitochondrial dysfunction in macrophages, and promoted integrins production and leukocyte migration in endothelial cells, have been observed (Figure 1C) [58]. Moreover, the GITR/GITRL complex affects tumor cell activity, mainly via increased expression of cell cycle regulators, immunomodulatory proteins, and apoptosis factors (e.g., transforming growth factor TGF-β, IL-10, TNF) (Figure 1C) [20].

## 3. Exploring the Role of GITR/GITRL Signaling in Liver Physiology and Disease

### 3.1. The Role of GITR/GITRL in Fostering Immune Tolerance

Tregs have a pivotal role in maintaining immune tolerance, and since they consistently express GITR, exploring GITR-mediated Tregs modulation is considered significant. In a recent study [59] investigating the role of GITR in the induction of antigen-specific Tregs in the liver-draining celiac lymph node (CLN) after oral antigen use, it was observed that ovalbumin use was associated with an enhanced Tregs activation in the CLN, characterized by CD25, GITR, CTLA-4 and CD103 co-expression. Additionally, ovalbumin-specific T cells from the CLN presented a reduced dependency on TGF-b for suppression, indicating GITR’s involvement in an alternative suppressive mechanism in this context. Overall, GITR appears to be crucial in promoting the regulatory phenotype of T cells in the CLN following oral antigen administration. Liao et al. demonstrated that GITRL aids effector T-cell and Treg proliferation. However, when they activated T cells and Tregs and subsequently treated them with GITRL, only Treg proliferation was noted, advocating that Tregs retained their suppressive ability. Interestingly, treatment with the agonistic GITR antibody DTA-1, rather than the natural ligand GITRL, led to a reduced percentage of Tregs, indicating that they do not function the same way in vivo [60]. Moreover, according to the Lin et al. study, GITR plays a crucial role in regulating the expression of the tumor necrosis factor 4-1BB on memory CD8^+^ T cells and is essential for the maintenance of CD8^+^ T-cell memory in vivo. Furthermore, the regulation of 4-1BB expression by GITR was proved significant in the pathological effects observed in unimmunized mice treated with anti-4-1BB agonists: mice lacking GITR were characterized by reduced cellular expansion, splenomegaly, and liver inflammation following anti-4-1BB treatment, indicating that GITR’s absence can mitigate the adverse effects of anti-4-1BB treatment [61].

Moreover, Cao et al. focused on gene replacement therapy, investigating specifically hepatic adeno-associated viral (AAV) gene transfer, which has shown promising results in inducing immune tolerance. Their study indicated that gene transfer is potentially associated with regulatory CD4^+^CD25^+^ Tregs induction, thus promoting immune tolerance. Specifically, the study identified an increase in CD4^+^CD25^+^GITR^+^ and FoxP3-expressing Tregs after hepatic gene transfer. These Tregs are essential for preventing the formation of antibodies against the therapeutic protein [62]. Towards this direction, Liao et al. investigated the role of GITRL in the expansion and function of Tregs [63]. Flt3L, a well-studied inducer of dendritic cells (DC) and macrophage proliferation, increased Treg numbers in wild-type (WT) mice. However, this expansion was significantly reduced in mice lacking GITRL, indicating its crucial role in peripheral Treg expansion. Furthermore, GITRL deficiency was associated with reduced numbers of specific DC subpopulations, particularly in the spleen, suggesting that GITRL affects the expansion and differentiation of certain DC subsets, which in turn impacts Treg expansion. In vitro studies revealed that GITR-L-deficient DCs were less effective in inducing the proliferation of antigen-specific Tregs and CD8+ T cells compared to WT DCs. After gene transfer using AAV8-OVA, GITRL-deficient mice showed reduced antigen-specific Tregs and an increased number of CD8+ T cells, suggesting GITRL plays a role in balancing Treg and CD8+ T cell responses [63].In conclusion, GITRL is essential for optimal Treg expansion and function, and its absence may lead to impaired Treg development and reduced immune regulatory capacity, especially in the context of gene therapy.

### 3.2. GITR Signaling in Infection and Immune Regulation

GITR signaling has emerged as a significant player in viral infections, with studies predominantly relying on animal models. A human study investigated GITR and PD1 levels in patients with HBV. It was observed that GITR expression decreased in CD8 Tregs, but not in CD4 Tregs, in patients with undetectable vs. those with detectable HbsAg levels 6 months post-infection. Additionally, GITR levels were lower in CD4 cells in the former group and marginally lower in CD8 cells, although this difference was not statistically significant. These observations suggest that reduced GITR levels may impair Treg suppressive function and enhance CD4 antiviral function [64]. Batista et al. also explored the role of GITR signaling in regulating the expression of CX3CR1 on CD4^+^ T cells during chronic viral infection with LCMV. GITR was identified as a crucial enhancer of CX3CR1 expression on CD4^+^ T cells post-infection. This increased expression of CX3CR1 was associated with a more differentiated Th1 effector CD4^+^ T cell phenotype. Interestingly, while CX3CR1 serves as a marker for these highly differentiated CD4^+^ T cells, its absence or deficiency did not impact the accumulation of CD4^+^ T cells in key organs such as the spleen, lung, or liver during the infection. This suggests that while GITR signaling influences CX3CR1 expression, it does not significantly affect the migration or accumulation of CD4^+^ T cells in these organs during chronic LCMV infection [65].

Interestingly, GITR signaling does not limit its influence on viral infections; it is also implicated in parasitic infections, as suggested by three studies investigating its role along with IL-10 expression. In detail, blocking IL-10 led to a significant reduction in parasite burden in a study on *S. mansoni* reinfection [66], while targeting GITR with an agonist anti-GITR antibody presented no effect on parasite levels or IFN-γ production in visceral leishmaniasis. However, when IL-10 was blocked, an increase in IFN-γ levels was observed. This evidence suggests that GITR may influence IL-10 signaling, affecting cytokine production or activating an unknown immune pathway. Treatment with sodium stibogluconate plus anti-GITR or anti-IL-10R antibodies reduced parasite burden in mice [67]. Walsch et al. demonstrated that both IL-4 and IL-10 are produced by CD25- Th2 cells and GITR was upregulated on CD25^+^ cells [68]. Moreover, in a study by Haque et al., GITR stimulation with an agonistic anti-GITR antibody reduced parasite burden in mice when administered 5 days after infection but not at the time of infection. This effect was dependent on the presence of CD4 cells and did not alter the number of Tregs in the liver [69]. Finally, the role of GITR was studied in murine filariasis by *Litomosoides sigmodontis* and schistosomiasis by *Schistosoma mansoni*. In *Litomosoides sigmodontis* -infected mice, GITR stimulation enhanced Th2 responses temporarily. However, in *Schistosoma mansoni* infection, blocking GITR led to a decrease in both Th1 and Th2 responses [70]. Overall, the studies described above support the potential role of GITR stimulation as an adjunctive treatment against parasites, although the transient nature of GITR’s effects may limit its therapeutic potential [66,67,68,69,70].

### 3.3. The Role of GITR Signaling in Liver Transplantation

Limited research has explored GITR’s role in liver transplantation. Zhang et al. investigated emodin’s impact on liver dendritic cells, revealing that it inhibits their maturation while boosting Treg numbers. These Tregs exhibited reduced GITR levels compared to controls, hinting at GITR’s therapeutic potential in liver transplantation [71]. Wei et al., using liver transplanted mouse models, proved the pronounced GITRL expression on allograft-derived Kupffer cells, with moderate expression in allograft liver cells versus isograft cells. This elevated TNF production in allograft cells was considered to be a result of GITRL activation, suggesting GITR could be involved in acute allograft rejection and potentially preventing graft failure [72]. Wang et al. explored the methylation status of GITR and GITRL genes as possible prognostic indicators for chronic rejection post-liver transplant. Preliminary results linked enhanced survival to increased methylation in the GITRL gene, while reduced methylation correlated with fewer acute rejections, underscoring the need for more comprehensive studies on GITRL and GITR methylation in liver transplant recipients [73].

### 3.4. Exploring the Role of GITR in Liver Cirrhosis

Recent studies have focused on examining the potential contribution of the GITR/GITRL system to the pathophysiology of liver cirrhosis and chronic hepatitis. In patients with liver cirrhosis, increased levels of GITR expression on both effector and regulatory CD4^+^ T cells were observed [74]. Liu et al. examined 27 chronic hepatitis B (CHB) patients with a clearance of hepatitis B surface antigen (HBsAg), who showed significant downregulation of GITR expression in total CD4^+^ T cells compared to HBsAg-positive controls [64]. Moreover, GITR levels were significantly decreased compared to HCs in a cohort study of 136 individuals investigating immune checkpoint regulator (ICs) levels in patients with CHB-related liver cirrhosis [75]. Conversely, a recent study contradicted those findings, showing elevated GITR levels as well as PD-1 and cytotoxic T lymphocyte-associated antigen 4 (CTLA-4) levels in treatment-naïve chronic HBV patients and individuals with HBV-related hepatic failure [76]. In the context of cirrhosis caused by hepatitis C virus (HCV) infection, in a retrospective study of 56 patients, GITR levels remained notably elevated more than two years after HCV clearance compared to healthy controls [77]. Furthermore, heightened levels of GITR prior to treatment were linked with a persistent elevation in hepatic steatosis index (HSI) following successful antiviral therapy in a multicenter retrospective study examining 62 individuals co-infected with HIV/HCV. This observation suggests a prospective role for pre-treatment GITR levels in the context of predicting the progress of liver disease [78].

### 3.5. Insights in the Potential Role of GITR in Autoimmune Hepatitis

Since it has been yet proven that GITR triggers resistance to tumors and viral infections, investigating its potential role in autoimmune liver disease (autoimmune hepatitis) is considered of great importance. However, solid evidence supporting the exact mechanisms by which GITR expression or targeting leads to autoimmune hepatitis pathogenetic events is still lacking. Recent studies have described its involvement in several autoimmune diseases (rheumatoid arthritis, Sjögren’s syndrome, autoimmune colitis, Hashimoto thyroiditis) pathogenesis and treatment, suggesting mechanisms by which it may be associated with autoimmune hepatitis regulation as well [79]. Specifically, GITR involvement in both innate and adaptive immunity is potentially correlated with the inflammatory activation of autoimmune hepatitis, suggesting that its pharmaceutical inhibition could be considered an effective treatment via inhibiting the activation of T-cells and inflammatory cells and, specifically, via sustaining the immunocompetence of Tregs, leading to autoimmune mechanisms blockade [80]. Currently, available results observed from research conducted in mouse models propose the promising role of GITR-Fc fusion protein on autoimmune microenvironment; however, future research is needed to establish these findings and translate them into clinical applications for the therapeutic evaluation of autoimmune hepatitis and other autoimmune diseases [81].

## 4. Investigating the Role of GITR/GITRL Signaling in HCC

### 4.1. GITR Expression in HCC

The composition of lymphocytes within normal liver tissue includes T cell subpopulations, consisting of T regulatory cells (Tregs), natural killer (NK) cells, and natural killer T (NKT) cells, all of which can potentially control tumor growth. Generally, within liver tumors, there is a dysregulation in the frequencies of these lymphocytes, indicating an immune-permissive environment that promotes tumor development [82]. We will focus herein on the number of Tregs within HCC tissues, which are considered the main expressors of GITR, as well as on GITR expression levels as observed from recent published pre-clinical studies [83,84].

Patients with HCC have been shown by Ormandy et al. to exhibit heightened numbers of CD4^+^CD25^+^ Tregs in their peripheral blood, accompanied by increased GITR expression on these Tregs, both within the tumor microenvironment and systemically [85]. In this study, patients were classified according to coexisting liver pathologies (hepatitis, cirrhosis, Child–Pugh score); however, no classification regarding histological type and stage was made [85]. Pedroza-Gonzalez et al. also documented significant findings regarding the immune landscape within liver tumors, particularly HCC and liver metastases from colorectal cancer (LM-CRC). They revealed a notable accumulation of functional Tregs within both HCC and LM-CRC tumors, suggesting their role in suppressing tumor-specific T-cell responses. Interestingly, even in HCC patients without prior liver disease, there was a significant accumulation of Tregs in the TME. Analysis of Treg markers revealed a more activated phenotype of intrahepatic Tregs compared to circulating Tregs, with tumor-infiltrating Tregs expressing higher levels of activation markers such as ICOS and GITR. Interestingly, the study demonstrated that engagement of GITR could counteract the suppressive effect of Tregs on Teffs within the HCC TME [82]. Those results are also supported by Zhang et al. research, supporting that upregulation of GITR coincides with enhanced proliferation of CD4^+^CD25^+^ Treg populations as well as concurrent upregulation of CTLA4 in patients with HCC [84,86]. Zhang et al. HCC classified patients via Tumor Node Metastasis (TNM) staging, specifically reporting eight patients at stage II, 19 patients at stage III and 22 patients at stage IV; however, no correlations regarding GITR expression and TNM stages were made [86]. Such enhancements in Treg function contribute to the suppression of anti-tumor immune responses against HCC antigens [86]. Regarding GITR’s expression levels, in a study examining the role of TIM-3, a member of T-cell immunoglobulin and mucin domain family, TIM-3 positive CD4 T cells within tumor tissues from various cancer types, including hepatocellular, cervical, colorectal, and ovarian carcinomas, have shown elevated expression levels of CD25, Foxp3, CTLA-4, and GITR when compared to their TIM-3 negative CD4 T cell counterparts [87]. Furthermore, a study examining 108 patients diagnosed with HCC concluded that the T cell population, when co-cultured with Huh7HCC cells, upregulates GITR. Huh7 culture supernatants appear to promote CD4^+^CD25^+^ T-cell proliferation, suggesting that tumor-related factors play a key role in CD4^+^CD25^+^ cell expansion and suppressor ability [83]. These findings combined suggest that increased Tregs and GITR expression levels are observed in HCC compared to normal tissue. Regarding the correlation between GITR expression levels and HCC patients’ survival, results reported by a study conducted by He et al. examined 421 HCC tissue samples showed a negative correlation between Tregs levels and GITR expression, also indicating that overall survival rates were significantly lower in patients with a higher GITRL expression compared [88]. Further research is needed in order to establish these findings in large patient cohorts and to correlate GITR/GITRL expression with tumor characteristics (e.g., HCC histologic type and stage).

Evidence regarding the comparison of GITR expression levels in HCC vs. other malignancies is ambiguous. Specifically, in a phase I trial evaluating the combination of stereotactic ablative radiotherapy (SABR) with ipilimumab in 35 patients diagnosed with various malignancies, individuals who underwent hepatic radiation exhibited elevated proportions of CD8^+^ cells expressing GITR compared to those receiving lung radiation [89]. Conversely, a bioinformatics analysis published about specific Treg-targeted antibodies reported that GITR did not meet the criteria for overexpression in liver carcinoma, in contrast to its overexpression observed in other malignancies such as squamous non-small cell lung cancer (NSCLC), renal cell carcinoma, breast infiltrating ductal carcinoma, ovarian serous carcinoma, and clear cell carcinoma [90].

### 4.2. GITR Involvement in Hepatocarcinogenesis Regulation

GITR’s involvement in immune responses against HCC has also been investigated lately. To begin with, Zhang et al. presented evidence in this direction in a study primarily exploring the role of CD4^+^CD25^+^ Tregs in modulating the immune response against HCC and investigating the potential therapeutic benefits of depleting these cells. Using various cell surface markers, including GITR, they distinguished Tregs from other activated T cells and observed an increase in Tregs expression levels in both peripheral blood and tumor tissue of HCC patients, providing a finding that correlates with the evidence previously described herein. The presence of Tregs was proved to inhibit the proliferative capacity of CD4^+^CD25^−^ effector T cells in a dose-dependent manner. The authors further aimed to promote the immune response against HCC by selectively reducing CD4^+^CD25^+^ Treg activity. Treg cell depletion was proved to enhance anti-tumor immune response in HCC patients, indicating potential therapeutic benefits. In conclusion, the study suggested that Tregs play a role in modulating the immune response against HCC, and using Treg depletion strategies along with established immunotherapeutic medication could significantly enhance anti-tumor immunity [84].

In the same direction, another study by Zhang et al. investigated the presence and characteristics of CD4^+^CD25^+^ Tregs in patients with chronic hepatitis B (CHB) and HCC [86]. Both CHB and HCC patients presented significantly higher CD4^+^CD25^+^ Tregs expression compared to healthy ones. Treg expression presented barely any difference between CHB and HCC patients. On the other hand, HCC patients presented different Tregs expressions regarding the presence of hepatitis B surface antigen (HBsAg), with those positive for HBsAg exhibiting higher levels of circulating Tregs compared to HBsAg-negative patients. Moreover, liver-infiltrating lymphocytes (LIL) from both CHB and HCC patients showed a notable increase in CD4^+^CD25^+^ Tregs compared to healthy controls. The authors suggested that Tregs play a significant role in modulating immune responses in CHB and HCC patients, being involved in tumor-specific immunity regulation and disease progression [86]. Thus, GITR/GITRL, as a modulator of Tregs could be considered a potential therapeutic target, modifying immune responses [91].

GITR’s role within HCC immune TME has also been studied regarding its correlation with CD40 expression, which is considered a marker of tumor growth. In detail, Murugaiyan et al. explored the role of CD40 expression levels on dendritic cells (DCs) in influencing tumor growth and regression [92]. Their study highlighted that the interaction between T-cell CD40 ligand (CD40-L) and antigen-presenting CD40 is important for T-cell activation and, furthermore, for tumor inhibition. It was proved that different levels of CD40 expression on DCs have the potential to suppress anti-tumor T-cell response. At low levels of CD40 expression, tumor growth is promoted, while higher levels induce tumor-regressing T-cell responses. This phenomenon was associated with differences in cytokine production and T-cell activation markers. Towards this direction, they investigated the role of GITR in the context of CD40 expression levels on DCs and their influence on tumor growth or regression [92]. Specifically, they showed that differential levels of CD40 expression on DCs were associated with varying GITR expression levels in T cells. High MHC-II/CD40-expressing DCs induced T cells expressing high levels of lymphocyte activation gene-3 (LAG-3) but low levels of GITR, suggesting a certain pattern of T cell activation. On the other hand, low MHC-II/CD40-expressing DCs induced T cells to express high levels of GITR and high levels of IL-4, indicating a different pattern of T cell activation. This finding implies that the expression levels of GITR on T cells could be modulated by the levels of CD40 expression on DCs, thus influencing the anti-tumor T cell response. Therefore, the study suggests that GITR expression is involved in the differential regulation of T cell responses, being negatively correlated with the expression of CD40, a marker of tumor growth decrease [92].

Moreover, GITR signaling has been studied along with leptin-mediated immune response as a part of HCC TME. Specifically, Wei et al. conducted a study in pre-clinical models investigating the role of leptin in modulating the immune response against HCC, particularly focusing on its effects on Tregs [93]. They observed that both hepatoma cells and HCC tissues produce higher levels of leptin compared to normal liver tissue. Leptin is found to up-regulate the expression of its receptor (LEPR) on T cells, especially Tregs, after HCC induction. Through in vitro experiments, it was demonstrated that leptin inhibited Treg activation and function by decreasing the expression of immunosuppressive molecules like TGF-β, IL-10, CTLA4 and GITR. Notably, the study demonstrated that GITR expression was reduced in Tregs treated with leptin, suggesting its involvement in leptin-mediated modulation of Treg activity. This reduction in GITR expression correlated with weakened Treg suppressive function, allowing for enhanced proliferation and cytotoxic activity of CD8^+^ T cells against hepatoma cells. Analogously, in a c12 HCC graft model in mice, leptin treatment was found to weaken the suppressive function of Tregs and enhance the cytotoxic activity of CD8^+^ T cells against hepatoma cells. This effect was observed when Tregs were pretreated with leptin before co-culturing with CD8^+^ T cells. Furthermore, when LEPR expression was silenced in Tregs, it led to increased expression of immunosuppressive markers (e.g., CTLA4 and GITR). This alteration in Treg function promoted HCC growth in the graft model, highlighting the crucial role of leptin signaling in regulating anti-tumor immunity in vivo [93]. Overall, the study uncovers a novel mechanism by which leptin enhances anti-HCC immunity by down-regulating Treg activity. The findings suggest that GITR, along with other immunosuppressive markers, plays a crucial role in mediating the effects of leptin on Treg function, ultimately influencing the balance between immune suppression and anti-tumor immunity in HCC [93].

### 4.3. GITR as an Individual Target in HCC

Treatment with soluble GITR has been investigated by one recent study. Tumor-derived Tregs presented a higher GITR expression compared to Tregs from non-cancerous tissues, suggesting that GITR could play a role as a Treg suppressor. Moreover, soluble GITR ligand (GITRL), when used at a low concentration of 10 μg/mL, has been associated with a significant reduction of tumor-derived Tregs proliferation and of effector CD4^+^ T cells cytokine production. Furthermore, a higher concentration of GITRL (20 µg/mL) was able to stimulate the proliferation of effector T cells in the absence of Tregs. These findings suggested that GITRL could relieve the suppression mediated by highly activated tumor-infiltrating Tregs, being a potential immunotherapeutic agent characterized by anti-tumor T cell activity within the tumor microenvironment [82,94].

Targeting GITR with an agonistic antibody as monotherapy has barely been studied. Pan et al. delved into the intricacies of immunotherapy using GITR agonistic antibody (DTA-1) in the context of HCC [95]. They began by elucidating the expression patterns of GITR across different immune cell types in the HCC microenvironment. Notably, tumor-infiltrating Tregs (Ti-Tregs) emerged as the primary expressors of GITR, particularly in AFP-positive HCC patients. This observation underscored the potential of Ti-Tregs as key targets for DTA-1 therapy. While DTA-1 effectively reduced Ti-Treg infiltration, it failed to activate CD8^+^ T cells, which is crucial for antitumor immunity. Furthermore, DTA-1 treatment induced an unexpected phenomenon: the polarization of macrophages towards an alternative M2 phenotype. This shift in macrophage polarization was associated with resistance to DTA-1 therapy in HCC. The study investigated further the mechanisms underlying DTA-1-induced M2 polarization. They uncovered that DTA-1 triggered a Th2 immune response characterized by elevated interleukin-4 (IL-4) secretion. This Th2 response, in turn, promoted M2 polarization of macrophages within the tumor microenvironment. Notably, DTA-1 did not directly influence macrophage polarization but rather indirectly through its effects on T cells, particularly Th2 responses. To overcome DTA-1 resistance mediated by M2 polarization, they explored potential therapeutic strategies. Combining DTA-1 with Toll-like receptor 4 (TLR4) agonists emerged as a promising approach [96]. TLR4 agonists effectively reversed M2 polarization induced by DTA-1, thereby enhancing its antitumor efficacy in HCC. In conclusion, this study provided an analysis of the interplay between immune cells, cytokine responses and therapeutic resistance mechanisms in HCC immunotherapy after using GITR agonistic antibody [95].

## 5. Anti-GITR Combination Treatment in Cancer Immunotherapy

Immune checkpoint inhibitors, including anti-PD-1, anti-PD-L1 and anti-CTLA-4, are considered a substantial advance in the therapeutic evaluation of solid tumors, including HCC [97,98,99,100]. Immune checkpoints include co-inhibitory molecules expressed by immune cells, preventing their functional activity [101]. Based on this mechanism, HCC avoids antitumor immune responses via the expression of the corresponding ligands both in tumor and normal tissue [102]. CTLA-4, PD-1, TIM-3, and LAG-3 are co-inhibitory receptors included in the current HCC therapeutic options, either commercially available or being investigated in clinical trials [100,101,102,103,104] and have been proven capable of eliminating tumor cells’ activity [105]. In the attempt to unravel the determinants of tumor resistance, the investigation has been directed towards TME, evaluating, at the same time, novel molecules that can be used as ancillary targets combined to establish HCC treatment options, enhancing their therapeutic value [106].

In detail, it has been proved in vivo that targeted therapy against PD-1/PD-L1 and agents targeting CTLA-1, despite presenting some points of convergence in their respective downstream pathways, have the potential to lead to different patterns of immune activation [107]. According to this evidence, attention has been shifted to the combination of two different immune checkpoint inhibitors, which is currently considered an effective strategy in the therapeutic evaluation of various types of solid tumors, including HCC [108,109,110]. The CheckMate 040 study, which included patients with advanced HCC, demonstrated notable tumor reduction in approximately 20% of individuals receiving anti-PD1 antibody (Nivolumab), indicating a positive objective response. However, despite these encouraging findings, over half of the patients did not exhibit a response to nivolumab treatment [111]. Consequently, there remains a pressing need for developing more efficacious immunotherapy medications and for establishing screening methods to identify suitable patients for HCC treatment.

As it has been described previously herein, GITR/GITRL upregulation in HCC promotes Treg-mediated immunosuppression, dampening anti-tumor responses [94]. Targeting GITR alongside CTLA-4 and PD-1/PD-L1 blockade demonstrates promising results in overcoming this suppression [112,113]. Moreover, Gal-9 inhibition combined with anti-GITR treatment enhances CD8^+^ T cell function, offering a promising strategy for improving HCC immunotherapy [114,115].

### 5.1. Anti-GITR Combined with Anti-PD-1/Anti-PD-L1

PD-1 represents a CD28 immune checkpoint molecule that negatively regulates T-cell activity by binding to its ligand PD-L1/2, blocking the stimulation signal of the T-cell receptor (TCR) [116]. PD-1 is expressed by activated T cells, NK cells, Tregs, MDSCs, monocytes and dendritic cells (DCs). Its ligand, PD-L1, is expressed by numerous stromal and tumor cells. PD1 and PD-L1 have been used as a prognostic and therapeutic marker for HCC [117]. During the last decade, immunotherapy targeting the PD-1/PD-L1 has shown promising results in HCC patients’ survival; however, for most patients with advanced HCC, targeting individually the PD-1/PD-L1 axis has been proved ineffective [101,118]. Thus, combination therapy may be a better option. In this context, combining antibodies against GITR and PD-1/PD-L1 axis theoretically arises. Studies investigating the combination of anti-PD-1/anti-PD-L1 and anti-GITR combination treatment have shown promising results, as described below.

To begin with, Van Beek et al. focused on investigating the potential of agonistic targeting of the co-stimulatory receptor GITR to enhance anti-tumor immune responses in HCC [97]. CD4^+^FoxP3^+^tumor-infiltrating lymphocytes exhibited high expression levels of GITR. Analysis of immune cell subsets within different tissue compartments revealed that while NK cells and NKT cells were less frequent in tumors compared to blood and tumor-free liver (TFL) tissues, T cells were more abundant in tumors. Specifically, CD4^+^FoxP3^+^ T cells were found to be accumulated in tumors, whereas CD8^+^ T cells were reduced. GITR expression was observed on various immune cell subsets, with the highest expression detected on CD4^+^FoxP3^+^TIL. Further characterization of CD4^+^ T-cell subsets revealed that activated Tregs, identified by FoxP3^hi^CD45RA^−^ phenotype, were highly abundant in tumors and displayed the highest levels of GITR expression. Activated Tregs were significantly increased in tumors compared to normal samples. Additionally, activated Tregs in tumors presented GITR and CD25 co-expression, indicating an activated phenotype. Proliferative activity was enhanced in both CD4^+^ and CD8^+^ TILs when targeting GITR with GITRL or an anti-GITR antibody. Specifically, CD8^+^ T-cell proliferation and granzyme B production were increased when using GITRL, while granzyme B and IFN-γ production in CD4^+^ and CD8^+^ T cells was increased by using anti-GITR antibodies. Moreover, when tumor antigen-specific responses were evaluated, GITR ligation resulted in significant enhancement of CD4^+^ and CD8^+^ TIL proliferation in response to tumor antigens. Combination therapy with PD-1 blockade (using nivolumab) and GITR ligation showed mixed effects on TIL responses (Figure 2). While there was no significant enhancement in proliferation compared to single treatments in the overall analysis, individual patient responses varied, with some patients showing benefits from the combination therapy (Figure 2). These findings combined suggest that targeting GITR can enhance the functionality of TIL in HCC, potentially offering a promising strategy for immunotherapy in this cancer type [97]. Overall, the study aimed to provide insights into the potential of GITR agonism as a therapeutic strategy for HCC and to explore its synergistic effects with PD1 blockade, with the ultimate goal of improving immunotherapy outcomes for HCC patients.

Analogously, Lenghans et al. explored the interplay between regulatory Tregs, immune checkpoint inhibition (ICI) and HCC [99]. They aimed to investigate whether Tregs might activate ICI pathways, especially PD-1, in HCC and elucidate the underlying cellular mechanisms. PD-1 inhibitors target the suppressive interaction between PD-1 on T cells and tumor ligands, yielding promising results in recent trials with nivolumab [108] and pembrolizumab for advanced HCC [119]. Notably, significant reductions in tumor size and objective response rates of up to 20% were observed. Interestingly, despite low expression of PD-1 ligands in tumor tissue, efficacy persisted, suggesting the involvement of alternative checkpoint pathways [111,119]. Peripheral blood samples from a total of 116 individuals were analyzed, including 50 HCC patients, 41 non-tumor-bearing liver disease controls and 25 healthy controls. Checkpoint molecules’ expression, specifically PD-1, PD-L1, CTLA-4, GITR and Tim-3 on Tregs, and their inhibitor-molecules’ expression, specifically IL-10, IL-35, TGF-beta, and galectin-9, were assessed via flow cytometry. Additionally, correlations between patient characteristics, Treg subsets, and checkpoint molecule expression were explored. Decreased expression levels of PD-1 and PD-L1 were observed in HCC tumors, despite CD8^+^ T cells infiltration, while Tregs from HCC samples showed high PD-1/PD-L1 expression levels, accompanied by increased secretion of inhibitory cytokines IL-10 and IL-35.

Furthermore, PD-1^+^ T cell levels were higher in HCC patients, while PD-L1^+^ Treg frequencies were increased in both HCC and non-tumor-bearing liver disease patients. The opposite observations occurred regarding the age of the patients and the frequency of Tregs expressing CTLA-4 and PD-L1 [99]. In conclusion, the study provides insights into the complex immunological dynamics of HCC, highlighting the role of Tregs and checkpoint inhibition, such as GITR blockade, in HCC progression. The findings suggest potential therapeutic strategies involving checkpoint inhibitors to enhance antitumoral activity in HCC. Overall, the study contributes to a deeper understanding of the immunological mechanisms underlying HCC pathogenesis and offers implications for the development of targeted immunotherapeutic approaches.

### 5.2. Anti-GITR Combined with Anti-CLTLA4

CTLA4 is expressed by activated T cells, blocking their activation, and is considered an effector molecule for Tregs [112]. CTLA-4 blockade has been proven to be an effective therapeutic option in patients with HCC. IBI310, an anti-CRLA4 monoclonal antibody in combination with the anti-PD-1 antibody Sintilimab, has demonstrated promising results in a Phase I study on patients with advanced HCC and is currently being investigated in a phase III study as a first-line treatment in advanced HCC patients, compared to Sorafenib [113]. Towards the direction of targeting specific molecules involved in Treg-mediated T- cell suppression, such as CTLA-4, anti-GITR treatment as monotherapy has been evaluated as a more effective enhancer of antitumor immune responses compared to its combination with CTLA-4 [98].

In detail, Pedroza-Gonzalez et al. aimed to investigate the efficacy of the two aforementioned immunotherapy strategies, GITR engagement and CTLA-4 blockade, either alone or in combination, in alleviating the immunosuppressive effects mediated by Tregs in liver tumors [98]. Firstly, they evaluated the impact of GITR engagement and CTLA-4 blockade individually. They used ex vivo isolated cells from patients’ tumors and found that treatment with a soluble form of GITRL or with blocking antibodies targeting CTLA-4 resulted in a reduction of the immunosuppressive activity exerted by Ti-Tregs. This reduction allowed for the restoration of effector T-cell proliferation and cytokine production, essential components of an effective antitumor immune response. Importantly, they subsequently explored the potential synergistic effects of combining GITR engagement and CTLA-4 blockade. They found that when low doses of both treatments were administered together, there was a more pronounced recovery of T cell function compared to either treatment alone. This combination therapy resulted in a stronger enhancement of effector T-cell proliferation and cytokine production, indicating a potentially greater efficacy in overcoming Ti-Treg-mediated immunosuppression (Figure 2). In summary, the study provides valuable insights into the potential of immunotherapy strategies, particularly GITR stimulation and CTLA-4 blockade, in overcoming Treg-mediated immunosuppression in HCC [98]. These findings need further investigation to be established and may ultimately contribute to the development of more effective treatment approaches for patients with HCC.

### 5.3. Anti-GITR Combined with Anti-GAL-9

In HCC, Galectin-9 (Gal-9) and GITR play crucial roles in regulating the TME and influencing the immune response against cancer [114,115]. Gal-9, primarily expressed in immune cells, particularly myeloid cells like dendritic cells and monocytes, exerts immunomodulatory functions within the TME [115]. Its expression is upregulated by interferon β (IFNβ) and, to a lesser extent, by IFN-γ [115]. Gal-9 acts as a mediator of immune response regulation, modulating T-cell function and promoting tumor immune evasion [116,117,118,119,120].

Yang et al. delved into the interplay between immune checkpoint proteins PD-1, Gal-9 and TIM-3 within the context of HCC immunology [121]. They revealed a novel role for PD-1 in modulating the Gal-9/TIM-3-mediated apoptotic pathway in T cells, shedding light on how HCC cells evade immune surveillance mechanisms [121]. Specifically, they demonstrated that PD-1 interacts with Gal-9 and TIM-3 to inhibit the apoptosis of T cells expressing both PD-1 and TIM-3, thereby promoting the persistence of exhausted T cells within the tumor. Exhausted T cells are a subset of immune cells found in tumors that exhibit reduced functionality and increased expression of inhibitory receptors such as PD-1 and TIM-3 [122]. Previous studies have highlighted the role of TIM-3 in inducing T cell apoptosis upon engagement with its ligand Gal-9 [122]. However, they documented that PD-1 competed with TIM-3 for binding to Gal-9, thereby attenuating TIM-3-mediated cell death signals and promoting T cell survival within the TME. Furthermore, research demonstrated that Gal-9 expression was upregulated by IFNβ and IFNγ. This suggests that targeting Gal-9 emerges as a potential strategy for cancer immunotherapy since inhibiting Gal-9 selectively expands certain T cell populations within tumors, including effector-like PD-1+TIM-3+ transitory T cells that play a crucial role in antitumor immunity. Moreover, they explored the therapeutic potential of combining Gal-9 inhibition with other treatment modalities. They showed that anti-Gal-9 treatment synergized with an agonist antibody targeting GITR, leading to enhanced expansion of CD8^+^ T cells and depletion of Treg cells within the HCC TME (Figure 3). This combination therapy demonstrated promising antitumor effects, suggesting a rationale for combining Gal-9 inhibition with anti-GITR that diminishes Treg cells and activates CD8^+^ T cells for improved cancer immunotherapy outcomes [121]. Overall, the study provides comprehensive insights into the complex interactions between immune checkpoint proteins and underscores the significance of targeting Gal-9 and GITR as a promising approach to overcome immune evasion mechanisms in HCC.

A brief summary of all the aforementioned studies’ findings regarding the role of GITR in HCC pathogenesis and prognosis is provided in Table 1.

## 6. Discussion

During the last decade, GITR has been investigated as a novel target that enhances the anti-tumor effects of the classic immunotherapeutic agents. DTA-1, previously referred to in this review, is the first monoclonal antibody developed to target GITR [123]. Various targeted agents were also tested, including antibodies with specific variations directing human GITR and antibodies with recombinant fusion proteins mimicking GITR [124,125]. Phase I clinical trials offer evidence supporting the safety of GITR antibodies [126,127]. The first-in-human trial (NCT-01239134) investigated TRX518, an IgG1 non-glycosylated monoclonal antibody that enhances NK cell activity and had previously presented promising results in preclinical trials [127]. TRX-518 was previously tested in vivo, showing effective results targeting GITR on CD4^+^ and CD8^+^ naïve and memory T cells, B cells, NK cells, invariant NK T cells, monocytes, and macrophages without depleting them. In the NCT-01239134 trial, GITR monotherapy depleted Tregs both in the periphery and in tumor tissue [127].

To date, nine GITR monospecific agonistic monoclonal antibodies (AMG-228, ASP1951, BMS-986156, GWN323, INCAGN1876, MK-1248, MK-4166, REGN6569, and TRX518) have been disclosed, studied as monotherapy or in combination with other chemotherapeutic choices, especially anti-PD-1 [128]. Of these, only TRX518 presented single-agent activity (1 responder with PD-1 and CTLA-4 refractory HCC) [127]. This evidence poses the question of whether evaluating GITR-targeting as a potential therapeutic agent in HCC treatment could be effective. GITR monotherapy is not generally considered effective in most types of tumors, although promising results have been observed using anti-GITR concomitant to other targeted drugs, particularly combined with PD-1 blockage [128]. Several clinical studies are ongoing, aiming to develop GITR-stimulating treatments [128]. Regardless of their structure (e.g., monospecific agonistic antibody, bispecific agonistic antibody, or fusion protein), the 7 GITR agonists (AMG-228, BMS-986156, GWN323, MEDI1873, MK-4166, MK-1248, and TRX518) studied as a monotherapy or in combination with PD-1 inhibitors or chemotherapeutic agents in patients with advanced solid tumors demonstrated no unusual safety signals [128]. Adverse events were rare and included fatigue, headache, decreased appetite, infusion-related reaction, nausea, abdominal pain, and pruritus, and were reported mainly after the combination of MK-4166 and pembrolizumab and after MEDI1873 monotherapy [126,129]. Only one patient with a dose-limiting adverse reaction ((bladder perforation in a urothelial patient with a neobladder) was reported after receiving GITR agonist (MK-4166) monotherapy [129,130]. No treatment-related deaths were observed [126,129]. It is worth noting that one study on mouse models reported anaphylactic events after repetitive doses of GITR agonist monoclonal antibody (DTA-1). However, in vivo results from human cohorts are needed to prove this finding [123].

Regarding HCC, the immune system represents an important way of controlling tumor progression, as the interaction between innate and adaptive immune systems enhances the use of effective antitumor immune surveillance [130]. Novel immunotherapeutic options for HCC have revolutionized HCC treatment, offering additional choices to patients with advanced disease. Specifically, continuous research on immunotherapeutic agents, including PD-1/PD-L1, TMB, ctDNA, microsatellite stability, DNA mismatch repair, neutrophil/lymphocyte ratio, cytokines, and cellular peripheral immune response has shown promising results, especially in patients with HCC resistant to classical chemotherapy [131,132,133]. Furthermore, characteristic examples of novel approaches are CAR T cells, which target specific antigens like glypican-3 (GPC3) T cell receptor (TCR)-based therapies, therapeutic vaccines aiming to enhance tumor-specific immune responses, HLA peptidomics and neoantigen identification, toll-like receptor, cGAS–STING RIGI or MDA5, and inflammasome agonists [131,132,133,134,135,136,137,138,139,140]. These new approaches come along with several levels of complexity, including overcoming chemoresistance and understanding the complex interaction when using combined immunotherapy drugs [131]. Research is continuously being conducted on how to appropriately sequence novel medicines for the best potential response, how to control toxicities, and how to develop indicators for monitoring patients in response and relapse states [131]. Several pieces of research are being conducted in large patient cohorts or pre-clinical models, including mouse models, human HCC tissue and cell lines, aiming to reveal combination treatment choices that are safe and effective, including immunotherapy enhancers [17,141,142].

Regarding published and ongoing clinical trials on advanced HCC therapeutic evaluation, clear evidence has not been observed, even in studies with similar inclusion criteria, such as those evaluating sorafenib or pembrolizumab (KEYNOTE-394) [143,144,145,146,147,148]. Furthermore, non-inferiority phase III trials aim to compare the efficacy of a new drug with the standard treatment. Such trials are crucial for drugs that may have additional benefits, like lower toxicity or cost. One critical aspect is determining the non-inferiority margin, which impacts how the trial results are interpreted and applied in clinical practice. For instance, the REFLECT and HIMALAYA trials aimed to show that new drugs, lenvatinib and durvalumab, respectively, were not inferior to Sorafenib in terms of OS [149,150,151]. They set a margin of 1.08 for the hazard ratio, meaning the new treatment could be up to 8% less effective than the standard without being considered worse. Choosing this margin involves balancing statistical precision with clinical practicality, as narrower margins require larger studies to prove non-inferiority, while wider margins may allow for smaller studies but could miss clinically relevant differences [151]. TIL therapy demonstrated feasibility in a phase I trial for HCC patients, while ongoing trials assess the efficacy of allogeneic NK cells [152,153]. In situ vaccines activate tumor-infiltrating antigen-presenting cells (APCs), while classic vaccines involve administering antigens or antigen-pulsed dendritic cells (DCs) [154,155]. While initial attempts faced efficacy challenges, recent advancements, like a multicenter trial in 2015, have shown improved PFS and OS for HCC patients treated with CIK cells after curative resection or ablation [156]. Tumor-infiltrating lymphocytes (TILs), NK cells, and chimeric antigen receptor (CAR) T cell therapy also show promise [155,157]. Finally, enhancing locoregional therapies, such as percutaneous ablation or intraarterial therapies, can induce immunogenic cell death or local delivery of immune-stimulating molecules [158,159]. In this context, GITR targeting, combined with other established immunotherapeutic drugs, has demonstrated promising immune effects in the expected tumor cell populations based on preclinical studies [108,109,110,111,112].

GITR presents a higher expression in HCC compared to normal tissue, a discrepancy partially explained by the increased number of Tregs in HCC tissue, which are the main GITR-expressing cells [83,84,85,86]. Moreover, its involvement in HCC TME has been demonstrated by a several number of studies [90,92,93]. Characteristically, it was observed that the expression levels of GITR on T cells could be modulated by the levels of CD40 expression on DCs, thus influencing the anti-tumor T cell response. Moreover, GITR expression has been proved to be involved in the differential regulation of T-cell responses, being negatively correlated with the expression of CD40, a marker of tumor growth decrease [91]. Treatment with soluble GITR is considered to alleviate the suppression mediated by Tregs, making it a potential enhancer of anti-tumor T cell activity within the TME [76,86]. The combination arms with anti-PD1 showed that a significant depletion of Tregs together with a high CD8^+^ TIL infiltration is required for clinical activity [80,84,87,88]. However, a significant difference between anti-GITR/anti-PD-1 combined therapy and anti-GITR monotherapy has not been demonstrated, while only pre-clinical activity is seen in these studies to draw meaningful conclusions. Combined anti-GITR/anti-CTLA-4, on the other hand, showed a significantly better response compared to anti-CTLA-4 monotherapy, which is considered a promising approach [98]. Moreover, anti-Gal-9/anti-GITR combination treatment leads to enhanced expansion of CD8^+^ T cells and depletion of Treg cells within the HCC TME [121]. While the exploration of GITR signaling as a therapeutic target in HCC shows promise, several limitations within existing studies warrant consideration. The majority of the studies described herein have primarily focused on animal models or human tissue, and the translation of findings to human patients may not always directly correlate due to biological differences. Moreover, the pre-clinical results of GITR targeting agents seem to be insufficient to induce significant clinical responses, while knowledge about further potential combinatorial approaches that could enhance the efficacy of these compounds is still barely investigated: further translational work is mandatory to translate the potential benefit of GITR agonists into a clinical level. The heterogeneity of HCC tumors among patients adds complexity, potentially influencing treatment responses. Moreover, the optimal timing, dosing, and combination strategies of GITR-targeted therapies remain areas requiring further investigation. Addressing these limitations through robust clinical studies and translational research efforts will be crucial for effectively translating the therapeutic potential of GITR modulation in the management of HCC.

## 7. Future Directions

Results from clinical and preclinical models, as described in this review, have demonstrated promising effects in several immune cell populations, with potential involvement in cancer immunotherapy; however, solid evidence for therapeutic effects in well-designed patient cohorts is still lacking. Continuously investigating GITR monospecific and bispecific agonists and co-stimulatory GITR ligands in different types of human neoplasms with a high incidence worldwide (e.g., breast and colorectal cancer) has improved our understanding of its potential future use in novel chemotherapy schemas [160,161]. Since GITR is expressed both on lymphocytes and tumor cells in the various types of neoplasms, future research should also include the involvement of its physiological expression in limiting tumor development. Considering its immunogenic effects on both types of cells, agonistic GITR-specific antibodies could be used to enhance the activity of weak anti-tumor immunogenic agents or vaccines. At the same time, efforts in the fields of translational research must be continued in order to improve pre-clinical models that have yet to show promising anti-tumor effects.

Specifically for HCC, which is characterized by extensive advances in immunotherapeutic options, further research into novel combination treatments should be kept under consideration in order to design new treatment regimens for advanced disease, in which immunotherapy could be considered as the treatment of choice. The results of in vivo studying DTA-1 or soluble GITR that were previously described herein suggest that anti-GITR antibodies produce co-stimulatory signals for responder CD4^+^ and CD8^+^ T cells [64]. It is currently clear that DTA-1 has promising antitumor properties in animal models; what is not clear yet is DTA-1′s relative effect on Tregs compared to T effector cells or endothelial cells. Recently, the idea of depleting an animal model’s Tregs and then enhancing the immune response with anti-GITR agents combined with co-stimulatory molecules (e.g., CD28, CD134, or TNFR proteins) has been reported [162]. Other authors suggest using GITR-specific agonists after developing antigen-specific T cells, which boost T cell response against tumor antigens, with concomitant use of molecules that block negative signaling (e.g., CTLA4, PD-1) [163]. Future research should focus on expanding the concurrent infiltration of GITR with various immune biomarkers, suggesting the promising effect of GITR as a part of combination immunotherapeutic options, an area that is already a part of several clinical trials on HCC patients. A multi-adjuvant approach must be applied in order to enhance immune response either in the form of multitargeted immunotherapy or in the form of novel technology of a tumor vaccine, as proposed by currently ongoing investigational projects [164].

## 8. Conclusions

In conclusion, the exploration of GITR signaling in HCC underscores its use as a promising therapeutic target, potentially being included in the avenues for precise therapeutic interventions arising from the interplay between the immune system and the HCC TME. Considering GITR’s pivotal role in regulating immune responses, the modulation of the GITR/GITRL signaling pathway emerges as a promising strategy for fortifying the immune system’s ability to combat and eradicate HCC cells. Targeting GITR signaling could be combined with established treatment options, in order to enhance the overall treatment efficacy. Specifically, using small molecule inhibitors combined with immune-modulating antibodies (including GITR agonists) could be the key to decreasing the immune threshold for anti-tumor effects, resulting in long-term anti-tumor immune effects. Further research is necessary to fully integrate this approach into clinical practice.

## Figures and Tables

**Figure 1 cancers-16-02609-f001:**
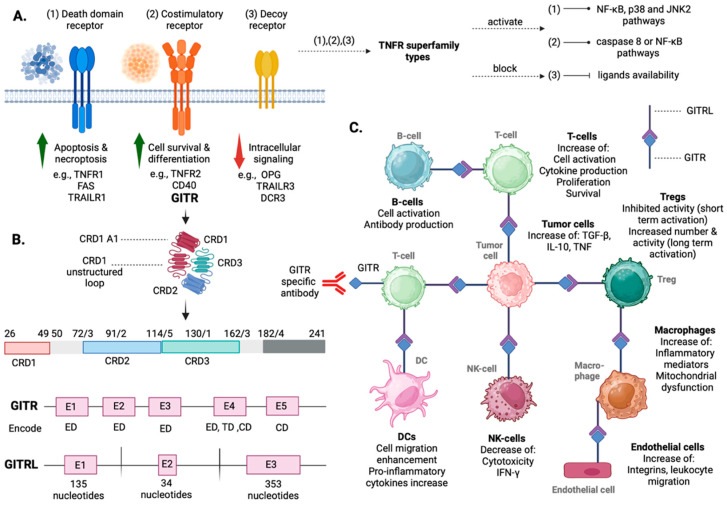
(**A**) TNFR superfamily types: costimulatory receptors, death domain receptors and decoy receptors, along with their basic promoting-inhibiting activity, (**B**) GITR and GITRL genes’ structure in exons (**C**) GITR/GITRL main effects in the various types of immune cells and their general pattern of tumorigenic activity. (CD: cytoplasmic domain; CRD: cysteine-rich domain; DC: dendritic cell, DCR3: decoy receptor 3; ED: extracellular domain; GITR: glucocorticoid-induced tumor necrosis factor receptor; NF-κB: nuclear factor kappa-light-chain enhancer; NK-cell: natural-killer cell; OPG: osteoprotegerin receptor; TD: transmembrane domain; Tregs: T regulatory cells; TNFR-TNFR1/2: tumor necrosis factor receptors; TRAIL: tumor necrosis factor-related apoptosis inducing-ligand).

**Figure 2 cancers-16-02609-f002:**
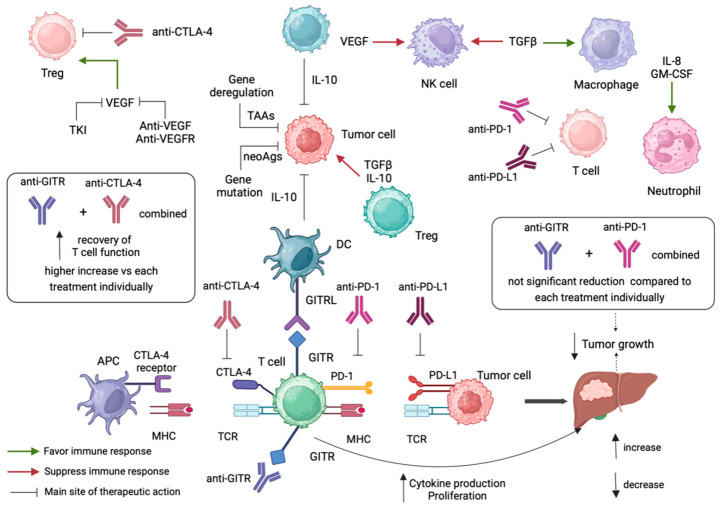
Schematic presentation of HCC immune network interactions and anti-GITR treatment in cancer immunotherapy, as monotherapy or combined with other antibodies (anti-PD-1, anti-CTLA-4). T cells, NK cells and DCs present a positive effect in immune tumor rejection, while Tregs, macrophages and neutrophils present a negative effect. In order to use targeted therapy, HCC cells should show antigen expression through gene mutations neoAgs or TAAs. Antibodies targeting CTLA-4 result in a reduction of immunosuppressive activity via increased recovery of T-cell function. This result is enhanced when anti-CTLA-4 therapy is combined with anti-GITR antibody. Anti-GITR therapy also results in an increase in cytokine reduction and proliferation, similar to anti-PD-1 and anti-PD-L1 targeting, decreasing tumor growth. However, combined anti-GITR and anti-PD-1 targeting has now yet shown a significant reduction of tumor growth compared to anti-GITR and anti-PD-1 monotherapy. (APC: antigen-presenting cell; CTLA-4: cytotoxic T lymphocyte-associated antigen 4; DC: dendritic cell; GITR: glucocorticoid-induced tumor necrosis factor receptor; GITRL: glucocorticoid-induced tumor necrosis factor receptor ligand; IL-8: interleukin 8; MHC: major histocompability complex; NK cell: natural killer cell; PD-1: programmed cell-death protein-1 PD-L1: programmed cell-death protein-1 ligand; TCR: T-cell receptor, TGFβ: tumor growth factor; VEGF: vascular endothelial growth factor; VEGFR: vascular endothelial growth factor receptor).

**Figure 3 cancers-16-02609-f003:**
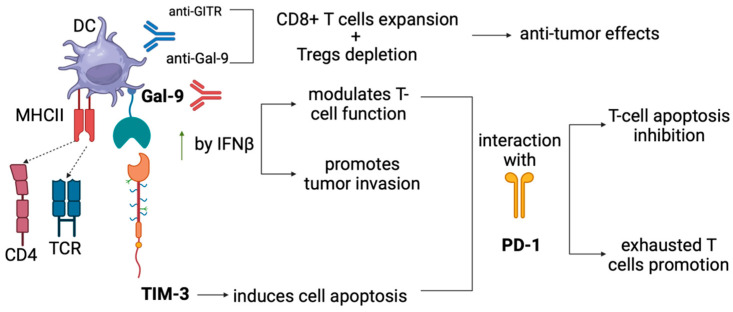
Schematic presentation of anti-Gal-9 anti-HCC effect and anti-Gal-9/anti-GITR combination treatment. PD-1 interacts with Gal-9 and TIM-3 to inhibit the apoptosis of T cells, expressing both PD-1 and TIM-3 and competes with TIM-3 for binding to Gal-9, attenuating TIM-3-mediated cell death signals and promoting T cell survival. Anti-Gal-9/anti-GITR combination treatment leads to enhanced expansion of CD8^+^ T cells and depletion of Treg cells within the HCC TME. (CD4: cytoplasmic domain 4; DC: dendritic cell; Gal-9: galectin-9, GITR: glucocorticoid-induced tumor necrosis factor receptor; MHCII: major histocompatibility complex 2; TCR: T cell receptor; TIM-3: T cell immunoglobin and mucin domain containing-3; Treg: T regulatory cell; PD-1: program cell death protein-1).

**Table 1 cancers-16-02609-t001:** Summary of studies evaluating the role of GITR in HCC pathogenesis and prognosis.

Authors (Year)	Primary Outcome	Secondary Outcome	Prognostic Value
Gonzalez et al. (2012) [82]	Treatment with GITRL increased the proliferation of CD4^+^/CD8^+^ T cells and cytokine production while it decreased Tregs suppression	Higher GITR expression was observed on Tregs in tumor tissue	Treatment of GITRL improves HCC prognosis, enhancing anti-tumor immunity (mainly via upregulation in CD4^+^/CD8^+^ and cytokines levels)
Cao et al. (2007) [83]	GITR expression was up-reregulated in T cells when co-cultured with HCC cells	Increased proliferation of CD4^+^/CD25^+^ T cells was observed when PBMCs and HCC cells were co-cultured	GITR upregulation improves HCC prognosis via enhanced immune response due to CD4^+^/CD25^+^ T cell proliferation
Zhang et al. (2010) [84]	Higher GITR expression on Tregs in tumor-infiltrating lymphocytes	Increased number of Tregs in HCC patients	GITR modulating Tregs depletion strategies may improve HCC prognosis
Ormandy et al. (2015) [85]	Higher GITR expression on CD4^+^/CD25^+^ T cells in HCC patients	Increased proliferation of CD4^+^/CD25^+^ T cells in HCC patients	CD4^+^/CD25^+^ T cell activity promotes tumor growth, and targeting them with immunotherapeutic agents may improve HCC prognosis
Zhang et al. (2009) [86]	Higher GITR expression on CD4^+^/CD25^+^ T cells and Tregs in tumor tissue	Increased proliferation of CD4^+^/CD25^+^ T cells in HCC patients	CD4^+^/CD25^+^ T cell activity promotes tumor growth, and targeting them with immunotherapeutic agents may improve HCC prognosis
He et al. (2022) [88]	The high GITRL expression group had a lower overall survival	Higher GITR and GITRL transcription in tumor tissue	GITRL is associated with lower survival rates, and blocking its activity may improve HCC prognosis
Cari et al. (2018) [90]	GITR did not satisfy the overexpression criterion in HCC	GITR was overexpressed in other types of malignancies	n/a
Murugaiyan et al. (2007) [92]	Lower GITR expression on T cells with higher CD40 expression	Higher CD40 expression on T cells promotes tumor growth	GITR expression is negatively correlated with negative HCC prognostic factors (CD40 expression) on Tregs
Wei et al. (2016) [93]	Lower GITR expression on Tregs after Leptin/LEPR inhibition	Upregulation of Leptin/LEPR on Tregs in HCC cells	Hepatoma cells enhance anti-HCC immune response via secreting leptin to decrease Tregs activity, promoting CD8^+^ T-cell response and improving HCC prognosis.
Pan et al. (2022) [95]	Higher GITR expression patients had higher survival prognoses. DTA-1 treatment decreased tumor size only when combined with anti-IL-4	Higher GITR expression on Tregs in tumor tissue	The increased GITR expression in tumor Tregs makes them a potential target for DTA-1 treatment, which may improve HCC prognosis.
Beek et al. (2019) [97]	The combination of GITRL and nivolumab increased the proliferation of CD4^+^/CD8^+^ T cells	Higher GITR expression on CD4^+^ and Treg in tumor tissue	GITRL/Nivolumab combination improves HCC prognosis, enhancing anti-tumor immunity via Tregs depletion
Gonzalez et al. (2015) [98]	GITRL and anti-CTLA-4 increased the proliferation of T cells and cytokine production	Higher GITR expression on Tregs in tumor tissue. GITRL decreased Tregs suppression	The increased GITR expression in tumor immune cells makes them a potential target for combination immunotherapy (targeting GITR and CTLA-4), with promising results in HCC prognosis.
Langhans et al. (2019) [99]	Anti-GITR did not decrease IFN-γ secretion in CD8^+^ T cells	GITR expression does not differ between patients and HCs	n/a

GITR: glucocorticoid-induced tumor necrosis factor receptor; GTRL: glucocorticoid-induced tumor necrosis factor receptor ligand; Tregs: regulatory T cells; HCC: hepatocellular carcinoma; PBMCs: peripheral blood mononuclear cells; CTLA-4: cytotoxic T lymphocyte-associated antigen 4; IFN-γ: interferon-gamma; IL-4: interleukin 4; LEPR: leptin receptor.

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
