# Peer review of "Exploring the Role of GITR/GITRL Signaling: From Liver Disease to Hepatocellular Carcinoma"

_cancers, 2024, doi:10.3390/cancers16142609_

Round 1

Reviewer 1 Report

Comments and Suggestions for Authors

Authors reviewed the role of GITR/GITRL in liver diseases and HCC.  Especially authors focused about immunotherapy with GITR for HCC. This review was comprehensively informative and well-written. In this review, efficacy in GITR-immunotherapy was described, but adverse effects were less addressed. Authors should describe adverse effects including possible ones according to the mechanism. They will be helpful for readers.    

Author Response

Authors reviewed the role of GITR/GITRL in liver diseases and HCC.  Especially authors focused about immunotherapy with GITR for HCC. This review was comprehensively informative and well-written. In this review, efficacy in GITR-immunotherapy was described, but adverse effects were less addressed. Authors should describe adverse effects including possible ones according to the mechanism. They will be helpful for readers.

We thank the reviewer for this suggestion. Adverse events after GITR-immunotherapy induction have not been extensively described in medical literature, however, we provide a summary of all the adverse events related to GITR immunotherapy, as they are observed from the completed or ongoing clinical trials using GITR monotherapy or combination treatment, in human and animal models (lines 739-752)   

Reviewer 2 Report

Comments and Suggestions for Authors

Here are some additional points to consider.

  1. HCCs have several histologic subtypes. It would be better to add GITR/GITRL expression differences in histologic subtypes of HCC.

  2. It would be better to describe GITR/GITRL expression according to HCC pathologic parameters (for example, histologic grade) and stage.

  3. The expression and role of GITR/GITRL in autoimmune hepatitis could have significant implications. It would be better to describe GITR/GITRL expression in autoimmune hepatitis.

     4.  Please add informations about prognosis of HCC in Table 1.

Comments on the Quality of English Language

Please check English grammar and spelllings.

For example,  Beek et AL (2019) -> Beek et al. (2019)

Author Response

Here are some additional points to consider.

  1. HCCs have several histologic subtypes. It would be better to add GITR/GITRL expression differences in histologic subtypes of HCC.

We thank the reviewer for this suggestion. As described (added) in text, patients’ classification regarding histological type and stage was not made in the currently available studies on GITR immunotherapy, and no correlations have been made between GITR/GITRL expression and tumor characteristics (e.g., HCC histologic type and stage). Patients were mostly classified according to coexisting liver pathologies and in one study the authors classified patients via Tumor Node Metastasis (TNM) staging, specifically reporting 8 patients at stage II, 19 patients at stage III and 22 patients at stage IV, however, no correlations regarding GITR expression and TNM stages were made.

This evidence is described in text, lines 362-364, 377-380 and 396-398.

  1. It would be better to describe GITR/GITRL expression according to HCC pathologic parameters (for example, histologic grade) and stage.

We thank the reviewer for this suggestion. Same as above, patients’ classification regarding histological type and stage was not made in the currently available studies on GITR immunotherapy, and no correlations have been made between GITR/GITRL expression and tumor characteristics (e.g., HCC histologic type and stage). Patients were mostly classified according to coexisting liver pathologies and in one study the authors classified patients via Tumor Node Metastasis (TNM) staging, specifically reporting 8 patients at stage II, 19 patients at stage III and 22 patients at stage IV, however, no correlations regarding GITR expression and TNM stages were made.

This evidence is described in text, lines 362-364, 377-380 and 396-398.

  1. The expression and role of GITR/GITRL in autoimmune hepatitis could have significant implications. It would be better to describe GITR/GITRL expression in autoimmune hepatitis.

We thank the reviewer for this suggestion. We added a distinct section describing the potential role of GITR/GITRL in autoimmune hepatitis, lines 333-349.

  1. Please add informations about prognosis of HCC in Table 1.

We thank the reviewer for this suggestion. A new section was added in table 1 about HCC prognosis and GITR-therapy/expression correlations (highlighted). 

Comments on the Quality of English Language

Please check English grammar and spelllings.

For example,  Beek et AL (2019) -> Beek et al. (2019)

We thank the reviewer for this suggestion. All these changes were made and are highlighted through the whole text.

Reviewer 3 Report

Comments and Suggestions for Authors

This narrative review about the role of GITR/GITRL signaling in liver diseases and hepatocellular carcinoma is comprehensive and detailed. The manuscript was significantly improved during the review process and main issues raised were appropriately addressed.

The overall quality of this narrative review is very good, and it is of interest for the researchers and clinicians specializing in chronic GI inflammatory diseases and GI cancers, including hepatocellular carcinoma. The manuscript summarizes new findings, discusses the most important molecular and cellular response pathways, also the role of GITR/GITRL signaling in immune response to the tumor and targeted immunotherapy.   

I don't have any further comments and support the publication of the paper in its current form.

Author Response

This narrative review about the role of GITR/GITRL signaling in liver diseases and hepatocellular carcinoma is comprehensive and detailed. The manuscript was significantly improved during the review process and main issues raised were appropriately addressed.

The overall quality of this narrative review is very good, and it is of interest for the researchers and clinicians specializing in chronic GI inflammatory diseases and GI cancers, including hepatocellular carcinoma. The manuscript summarizes new findings, discusses the most important molecular and cellular response pathways, also the role of GITR/GITRL signaling in immune response to the tumor and targeted immunotherapy.   

I don't have any further comments and support the publication of the paper in its current form.

We thank the reviewer for these comments, they are highly appreciated.

Reviewer 4 Report

Comments and Suggestions for Authors

The authors provide a comprehensive review on GITR signaling and its role in hepatocellular carcinoma and its potential targeting as a way to treat this carcinoma. It is clear that the authors have performed an extensive literature review and the references are all appropriate to the paper. This paper will be useful to researchers looking for a comprehensive summary of GITR's role in cancer. The paper only needs minor revisions as listed below.

1) The species names like S. mansoni and L. sigmodontis should be italicized. 

2) In Table 1, "et Al" should be changed to "et al." Moreover, this table is a little hard to read due to the lack of lines or a grid separating the different studies. Could the authors please do something to improve the readability of this table?

3) In line 463, "10 ug/m" should be corrected to "10 ug/mL"

4) In citing references, some spaces are incorrect like in line 741 after the REFLECT and HIMALAYA studies, there should be a space between those words and the reference in square brackets.

5) The main revision that needs to be made is the Conclusion-Future Directions. This should be split into two parts, the Conclusion and Future Directions. Please expand both sections so that the reader easily read a conclusive summary of the article and also have a good sense of the future directions of GITR in cancer therapy.

Author Response

The authors provide a comprehensive review on GITR signaling and its role in hepatocellular carcinoma and its potential targeting as a way to treat this carcinoma. It is clear that the authors have performed an extensive literature review and the references are all appropriate to the paper. This paper will be useful to researchers looking for a comprehensive summary of GITR's role in cancer. The paper only needs minor revisions as listed below.

  • The species names like S. mansoni and L. sigmodontis should be italicized. 

We thank the reviewer for this suggestion. This change was made (lines 288-296).

  • In Table 1, "et Al" should be changed to "et al." Moreover, this table is a little hard to read due to the lack of lines or a grid separating the different studies. Could the authors please do something to improve the readability of this table?

We thank the reviewer for this suggestion. These changes were made in Table 1 (highlighted).

  • In line 463, "10 ug/m" should be corrected to "10 ug/mL"

We thank the reviewer for this suggestion. These changes were made (highlighted).

  • In citing references, some spaces are incorrect like in line 741 after the REFLECT and HIMALAYA studies, there should be a space between those words and the reference in square brackets.

We thank the reviewer for this suggestion. These changes were made (highlighted).

  • The main revision that needs to be made is the Conclusion-Future Directions. This should be split into two parts, the Conclusion and Future Directions. Please expand both sections so that the reader easily read a conclusive summary of the article and also have a good sense of the future directions of GITR in cancer therapy.

We thank the reviewer for this suggestion. A distinct section for Future Directions was formed, lines 835-868, and Conclusions section was updated.
